Distribution and population structure in the naked goby Gobiosoma bosc (Perciformes: Gobiidae) along a salinity gradient in two western Atlantic estuaries

Moore Christopher S. moorech16@students.ecu.edu
Ruocchio Matthew J.
Blakeslee April M.H.
Biology Department, East Carolina University , Greenville , NC , United States of America
Toonen Robert
Electronic publication date: 2018 Aug 7
Publication date: 2018
Volume: 6
Electronic Location ID: e5380
Received 2018 Mar 15; Accepted 2018 Jul 11
Copyright: ©2018 Moore et al.
Copyright year: 2018
Copyright holder: Moore et al.
License: This is an open access article distributed under the terms of the Creative Commons Attribution License, which permits unrestricted use, distribution, reproduction and adaptation in any medium and for any purpose provided that it is properly attributed. For attribution, the original author(s), title, publication source (PeerJ) and either DOI or URL of the article must be cited.
License URL: https://creativecommons.org/licenses/by/4.0/

Keywords: Biogeography, Gene flow, Dispersal, North Carolina, Habitat mosaic, Demersal, Genetic diversity, Local, Regional

Funding: East Carolina University Oak Ridge Associated Universities This work was supported by East Carolina University (REDE and Thomas Harriot College of Arts and Sciences) and the Oak Ridge Associated Universities Ralph E. Powe Junior Faculty Enhancement Award to Dr. April Blakeslee. The funders had no role in study design, data collection and analysis, decision to publish, or preparation of the manuscript.

==============================
Many species of fish produce larvae that undergo a prolonged dispersal phase. However, evidence from a number of recent studies on demersal fishes suggests that the dispersal of propagules may not be strongly correlated with gene flow. Instead, other factors like larval behavior and the availability of preferred settlement habitat may be more important to maintaining population structure. We used an ecologically important benthic fish species, Gobiosoma bosc (naked goby), to investigate local and regional scale population structure and gene flow along a salinity gradient (∼3 ppt to ∼18 ppt) in two North Carolina estuaries. G. bosc is an abundant and geographically widespread species that requires complex but patchy microhabitat (e.g. oyster reefs, rubble, woody debris) for reproduction and refuge. We sequenced 155 fish from 10 sites, using a common barcoding gene (COI). We also included recent sequence data from GenBank to determine how North Carolina populations fit into the larger biogeographic understanding of this species. In North Carolina, we found a significant amount of gene flow within and between estuaries. Our analysis also showed high predicted genetic diversity based upon a large number of rare haplotypes found within many of our sampled populations. Moreover, we detected a number of new haplotypes in North Carolina that had not yet been observed in prior work. Sampling along a salinity gradient did not reveal any significant positive or negative correlations between salinity and genetic diversity, nor the proportion of singleton haplotypes, with the exception of a positive correlation between salinity standard deviation and genetic diversity. We also found evidence that an introduced European population of naked gobies may have originated from an Atlantic source population. Altogether, this system offers a compelling way to evaluate whether factors other than dispersal per se mediate recruitment in an estuarine-dependent species of fish with a larval dispersal phase. It also demonstrates the importance of exploring both smaller and larger scale population structure in marine organisms to better understand local and regional patterns of population connectivity and gene flow.

Introduction

The Gobiidae is the largest family of marine fishes, with nearly 2,000 species described worldwide, an estimated 320 of which are found in the Americas (Van Tassell, 2011). Cryptic by nature, gobies are small (<70 mm standard length) benthic fishes that are sedentary as adults. Most species inhabit tropical and sub-tropical regions (Thacker, 2011); however, some taxa, including members of the genus Gobiosoma, are also found in temperate latitudes in the western Atlantic. Although common across a broad range of estuarine habitats and salinity gradients, the Gobiosoma remain relatively understudied (Carle & Hastings, 1982; Van Tassell, 2011).

One member of this genus, the naked goby Gobiosoma bosc (Lacepede, 1800), is a geographically widespread species—ranging from Connecticut to Campeche, Mexico—and the most commonly encountered gobioid fish in estuaries of the southeastern United States (Dawson, 1966; Ross & Rohde, 2004). This small goby (<60 mm SL) is estuarine-dependent (Able & Fahay, 1988; Able & Fahay, 2010) and prefers structured habitat (e.g., oyster reefs, woody debris). G. bosc is most abundant in waters of low-to-moderate salinity (Dahlberg & Conyers, 1972), though it may also occupy sub-tidal mud flats or the shallow margins of marsh creeks (Miller & Guillory, 1980; Peterson & Ross, 1991; Hendon, Peterson & Comyns, 2000). Naked gobies function as an important trophic link between benthic and pelagic communities (Markle & Grant, 1970; Breitburg, Palmer & Loher, 1995; Breitburg, 1999).

Early studies characterizing the life history and distribution of G. bosc focused on its reproductive biology and the distribution and abundance of larvae and adults as a function of salinity (Massmann, Norcross & Joseph, 1963; Dawson, 1966; Dahlberg & Conyers, 1972; Crabtree & Middaugh, 1982; Shenker et al., 1983; Conn, 1989). Planktonic G. bosc larvae settle out and become part of the benthos at an approximate total length of 7–12 mm (Borges et al., 2011). Prior to settlement, larvae aggregate in low flow areas on the downcurrent side of oyster reefs (Breitburg, Palmer & Loher, 1995), which serve as the preferred habitat for juveniles and adults and are also integral to goby refuge and reproduction (Nelson, 1928; Massmann, Norcross & Joseph, 1963; Dahlberg & Conyers, 1972; Shenker et al., 1983; Lederhouse, 2009). In particular, adhesive egg masses are attached to the underside of oyster shells (e.g., Dahlberg & Conyers, 1972; Crabtree & Middaugh, 1982), and eggs hatch after approximately 1–2 weeks (Nero, 1976). Early-stage larvae are subject to passive dispersal processes, while larvae in more advanced stages of development are capable of positive rheotaxis and have been observed to aggregate around oyster reefs, or other structured habitat like rubble or artificial structures, prior to settlement (Breitburg, 1989; Breitburg, 1991; Breitburg, Palmer & Loher, 1995).

Due to its dependence on structured habitat for refuge and reproduction, spatially-mediated patterns of G. bosc larval settlement may determine local population densities throughout estuarine habitats (Breitburg, Palmer & Loher, 1995). Oyster reefs in particular form a complex spatial mosaic in estuaries, along with mudflats, submerged aquatic vegetation (SAV), and emergent vegetation (e.g., Bell, McCoy & Mushinsky, 1991; Skilleter & Loneragan, 2003; Gain et al., 2017), and this mosaic will in turn strongly influence goby abundance and distributional patterns. Further, larval G. bosc have been reported to migrate towards lower salinity regions in estuaries (e.g., Shenker et al., 1983); however, the home range of adult naked gobies and their ability to disperse downstream remain unknown (Ross & Rohde, 2004). In sedentary fishes like G. bosc, it is assumed that dispersal promotes connectivity between populations that are otherwise isolated within such habitat mosaics (Leis & McCormick, 2002; Shanks, 2009). In other words, long-distance dispersal of larvae over many kilometers would be expected to result in a relatively uniform population structure, especially at smaller spatial scales (Palumbi, 1994; Shulman & Bermingham, 1995; Mora & Sale, 2002; Palumbi & Warner, 2003). However, evidence from demersal tropical and Antarctic fishes suggests that self-recruitment and local adaptation can persist despite widespread gene flow (Ohman et al., 1998; Robertson, 2001; Taylor & Hellberg, 2003; Moody et al., 2015). This has led many to conclude that larval behavior and the availability of settlement habitat are more important predictors of population structure than dispersal alone (Shanks, 2009; Riginos et al., 2011; Kohn & Clements, 2011).

To date, few studies have attempted to quantify population connectivity in temperate fishes (e.g., Kohn & Clements, 2011) like G. bosc. However, a recent study on the population genetics of naked gobies (Mila et al., 2017) has provided some understanding of the large-scale phylogeographic differences throughout much of the species’ range. This investigation found that G. bosc is strongly structured based on geography, particularly between Gulf vs Atlantic populations, but within those major regions, gobies are further subdivided into subclades based upon geography (e.g., East and West of the Apalachicola River along the Florida panhandle), due to biogeographic breaks that inhibit gene flow. The Mila et al. (2017) study is therefore instrumental in understanding the broader phylogeography of G. bosc in North America. To date, however, there have been no studies investigating genetic diversity at smaller scales (i.e., at the estuary level). This is particularly relevant given the importance of suitable habitat in structuring populations of naked gobies, which are otherwise isolated from one another as adults within a broader habitat matrix.

In our study, we collected G. bosc along a salinity gradient from two North Carolina estuaries in order to quantify population connectivity within and between estuaries, and also to better understand the role of habitat and salinity in mediating gene flow and local adaptation. While naked gobies have a broad salinity tolerance (<1.0 ppt–32 ppt), they have been noted as most abundant in salinities below 24 ppt (Massmann, Norcross & Joseph, 1963; Shenker et al., 1983), and in North Carolina, we have observed adults to be most abundant in low-to-mid mesohaline habitats (5.0 ppt–16.0 ppt) (personal observations). We therefore hypothesized that populations in oligohaline (<5.0 ppt) and polyhaline (>24.0 ppt) salinities may be more subject to processes that lower genetic diversity, like drift.

To quantify population structure in these estuaries, we used the common mitochondrial barcoding gene Cytochrome Oxidase I (COI), which has been used as a tool for population genetics studies across multiple diverse taxa over the past 20+ years (NCBI, 2014; Barcode of Life, 2018), and specifically has been proven effective for understanding the genetic diversity of marine fishes (e.g., Ward et al., 2005). Furthermore, the use of this marker allowed us to explore our data in combination with other recent population genetics datasets for this species, like the Mila et al. (2017) study, and an additional goby phylogenetics study encompassing multiple goby species (e.g., Van Tassell et al., 2015). It is also noteworthy that although the comprehensive Mila et al. (2017) study sampled G. bosc throughout most of its range, it did not include samples from North Carolina, a state which features the second largest estuarine system in the USA (APNEP, 2018). Our study therefore provides a greater understanding of the importance of large estuaries in the life history G. bosc—particularly related to patterns of dispersal and connectivity among isolated populations of these and similar fishes. This work could therefore have important implications for the management of marine protected areas (MPAs), or other conserved habitats that are spatially segregated but connected by gene flow.

Materials and Methods

Study location

Gobiosoma bosc fish were sampled along a salient gradient spanning the Pamlico and Neuse River estuaries in the North Carolina coastal plain (Fig. 1; Table 1). Both the Pamlico and Neuse are shallow (average depth 1–3 m), microtidal (<1 ft. tidal range), oligohaline-mesohaline (0.5‰ to 18‰) estuaries that combine to form Pamlico Sound: the largest lagoonal estuary in the United States (Bales & Nelson, 1988). The Pamlico River estuary is a continuation of the freshwater Tar River, and it flows approximately 65 km from the town of Washington, NC to Pamlico Sound. The Neuse River estuary begins in New Bern, NC, and empties into Pamlico Sound. For comparison to samples collected from sites along the Pamlico and Neuse rivers, fish were also collected from Hoop Pole Creek Nature Preserve (HPC), which is located along Bogue Sound in Atlantic Beach, NC (Fig. 1) and is part of the Pamlico Sound.

Figure 1 Sample locations and their associated haplotype frequencies.

Within the eastern part of North Carolina (USA), Gobiosoma bosc were collected from 5 sites on the Pamlico River (3–10 ppt), and 5 sites along the Neuse river (3–15 ppt), as well as a single site from Bogue Sound (30 ppt), which feeds into the Pamlico Sound. Haplotype frequencies are shown for sites along the Pamlico (GSC, Goose Creek; MLC, Mallard Creek; NCL, North Creek Landing; WRC, Wright’s Creek), the Neuse (FSL, Fisher’s Landing; CQC, Cahooque Creek; MTP, Matthew’s Point; POC, Pin Oak Court; CDI, Cedar Island), and Bogue Sound (HPC, Hoop Pole Creek). Colored pie pieces represent shared haplotypes within North Carolina, and black pie pieces represent the collective proportion of singleton haplotypes per population (see Table 1 and Table S1 for more information). Base map layers courtesy of d-maps (http://d-maps.com/).

Table 1 Site information and sample data.

This table includes the following pertinent information related to the North Carolina populations included in the study: Site abbreviation (abbrev); Site name (including city); Location (estuary); Mean salinity (ppt) across four time points; the standard deviation of salinity (ppt) across four time points; classification of salinity into low (<5 ppt), medium (6–20 ppt), and high (21–32 ppt) categories; the total number of samples per site; the total number of haplotypes observed per site; the expected number of haplotypes per site based on rarefaction analysis; the number of predicted samples needed to reach the asymptote based on rarefaction analysis; the genetic diversity calculated per site; and the proportion of singleton haplotypes (out of all detected haplotypes) per site. Population level genetic diversity was not calculated for HPC.

Site abbrev	Site name	Location	Mean salinity (ppt)	Standard deviation salinity	Salinity classification	Number of samples	Total number of haplotypes	Expected # haplotypes	# samples to reach asymptote	Genetic diversity	Proportion of singleton haplotypes	
GSC	Goose Creek, Washington, NC	Pamlico	3.08	1.97	Low	34	8	22.53 (±2.36)	550	0.46	0.50	
MLC	Mallard Creek, Washington, NC	Pamlico	3.62	1.76	Low	19	5	9.65 (±1.33)	65	0.39	0.40	
NCL	North Creek Landing, Belhaven, NC	Pamlico	8.21	1.45	Medium	21	6	13.40 (±1.23)	80	0.43	0.50	
WRC	Wright’s Creek, Belhaven, NC	Pamlico	9.55	0.93	Medium	14	2	2.44 (±0.14)	15	0.14	0.00	
FSL	Fisher’s Landing, New Bern, NC	Neuse	2.91	1.78	Low	7	2	2.57 (±0.14)	8	0.29	0.50	
CQC	Cahooque Creek, Havelock, NC	Neuse	3.64	3.03	Low	15	8	29.94 (±2.91)	365	0.76	0.75	
MTP	Matthew’s Point Marina, Havelock, NC	Neuse	10.33	4.15	Medium	16	5	9.62 (±1.27)	55	0.45	0.60	
POC	Pin Oak Court, Merrimon, NC	Neuse	9.65	4.17	Medium	4	3	4.36 (±0.21)	12	0.83	0.33	
CDI	Cedar Island, NC	Neuse	14.21	4.09	Medium	17	6	13.61 (±1.70)	70	0.51	0.50	
HPC	Hoop Pole Creek, Atlantic Beach, NC	Bogue sound	30	n/a	High	8	3	3.57 (±0.12)	30	n/a	0.33	

Specimen collection

From February to July 2017, naked gobies were sampled (n = 155) from nine locations along the Pamlico and Neuse River estuaries, in addition to the site at Hoop Pole Creek (Fig. 1; Table 1) (North Carolina Division of Marine Fisheries Scientific or Educational Permit Number 706671). Fish were collected using passive collecting devices: small plastic milk crates (19.05 × 22.10 × 15.75 cm) filled with approximately 1.7 kg of autoclaved oyster shell. This technique is modeled on the successful methodology employed by the Smithsonian Environmental Research Center (SERC) for the past twenty years, (e.g., Roche & Torchin, 2007), and within our lab for the past 5 years. Although G. bosc and other organisms can freely move inside and outside the crates, they are attracted to the complex three-dimensional habitat that the shell provides. This is analogous to the collecting strategy used by D’Aguillo, Harold & Darden (2014), who employed habitat traps (i.e., “shell-rubble trays”: 0.8 m2 plastic trays covered with 0.6 cm mesh netting and filled with oyster shell) to sample specifically for naked gobies.

Two replicate crates were deployed at each sample location, making for a total of ten collecting units on the Neuse River and eight on the Pamlico River. Crates were zip-tied to 0.75 m wooden stakes secured in the nearshore subtidal zone, or deployed from fixed or floating docks using rope. Crates on both rivers were checked every 6–8 weeks, and the contents sorted using a large sieve (55.9 × 55.9 × 12.7 cm) with 2 mm mesh. A maximum of ten naked gobies were collected from each sample site during each sampling event. In order to minimize selection bias, fish from both crates were pooled together, and ten individuals were randomly selected from a grid divided into four quadrants. Only sexually mature adults were used in this study. Therefore, all fish less than 20 mm SL (i.e., Dahlberg & Conyers, 1972) were released at a minimum distance of 50 m from the collecting location. Fish were then live-transported to East Carolina University (ECU) and housed in aerated plastic aquaria (36.83 × 22.35 × 24.38 cm) at a salinity approximating that of the collecting location. All collection and housing protocols were approved by ECU IACUC: AUP #D346. Because our sample sites were located along a salinity gradient, salinity data were collected at each site using a handheld YSI (YSI Inc., Yellow Springs, OH). These point measurements were averaged across all sampling events (n = 4) to provide an average and standard deviation salinity value for each site that could be regressed against genetic (haplotype) diversity at each sample location.

DNA sequencing and analysis

Naked gobies were dissected as part of an unrelated study assessing parasite diversity in these fish, and thus sampled fish were not released following capture. Once dissected, white muscle tissue was saved from each individual and immediately preserved at −20 °C for later DNA extraction. Genomic DNA was isolated from tissue samples using proteinase K/SDS digestion, chloroform extraction, and ethanol precipitation (Kocher et al., 1989). Cytochrome Oxidase I (COI) PCR primers were designed based on sequence data from Van Tassell et al. (2015). These newly designed primers were: GOBY COI F: GCACCGCTTTAAGCCTTTTA and GOBY COI R: TGGTGTTGAGGTTTCGGTCT. The PCR profile is as follows: 95 °C for 2-min; 30 cycles of 95 °C for 30 s, 55 °C for 30 s, and 72 °C for 60 s; and 72 °C for 5-min (Blakeslee et al., 2017). PCR amplicons were purified using ExoSAP-IT™ (ThermoFisher Scientific). Samples were then sent for Sanger sequencing to Macrogen USA (Rockville, MD).

Sequences were manually cleaned, inspected for ambiguities, and aligned without gaps to a reference sequence in GenBank (accession #: KM077829.1) using Geneious10.1.2 (Biomatters Ltd., Auckland, New Zealand). This resulted in a 530 base-pair fragment of the COI gene across all samples (n = 155 individuals from North Carolina; accession numbers: MH680722–MH680751; Table S1). Sixty-six additional G. bosc sequences were located on GenBank following a BLAST search (https://blast.ncbi.nlm.nih.gov/Blast.cgi) with 100% coverage of the 530 base-pair COI fragment. These included 57 sequences (Popset: 1229627432; accession #s: MF168978–MF169034) from a recent and comprehensive population genetics investigation of G. bosc in several North American populations, including New York, Virginia, South Carolina, Atlantic Florida, Gulf Florida, Louisiana, Mississippi, and Texas (Mila et al., 2017). An additional nine sequences came from a study by Van Tassell et al. (2015): five of these were from a non-native population (Germany) and four were from the USA (Florida and the Gulf of Mexico) (accession #s: KT278516, KT278523, KT278535, KT278549, KT278552, KM077826, KM077829, KT278549, and KM077828).

Our new sequences were combined with the sequences from GenBank and aligned using Geneious 10.1.2. Sequences were then collapsed into haplotypes using TCS1.21 (Clement et al., 2002) (Table S1). The Mila et al. (2017) popset (#1229627432) from GenBank included an incidence-based understanding of the haplotypes found across the Atlantic and Gulf of Mexico. In other words, this popset contained no information on haplotype frequencies per population. On the other hand, our North Carolina dataset included frequency data that were explored between and among North Carolina populations and estuaries. As a result, two separate analyses were performed: (1) an investigation for North Carolina populations only, and (2) a geographic, incidence-based investigation of all haplotypes (including our new North Carolina ones) detected across the geographic range. This latter analysis determined how naked gobies in North Carolina fit into the larger biogeographic picture.

Genetic analyses

For our first analysis focused on North Carolina, we estimated the hierarchical analysis of molecular variance (AMOVA) using ARLEQUIN311 (Excoffier, Laval & Schneider, 2005). Resulting fixation indices helped pinpoint whether there was divergence among populations and between the two major estuaries in our study. Pairwise ϕSTs were calculated using ARLEQUIN (Table S2) and visualized in a non-metric multidimensional scaling analysis (using PRIMER 7.0.13 (Primer-e, Quest Research Limited, Auckland, New Zealand)) to look for spatial patterns among populations. We used PopArt (http://popart.otago.ac.nz/index.shtml) to graphically create haplotype networks.

We also used ARLEQUIN to obtain genetic diversity values for each population to investigate whether there was any effect of salinity on haplotype diversity in the North Carolina populations. For this salinity analysis, we included sites that were sampled from the two rivers (n = 9 sites: four in the Pamlico River and five in the Neuse River). We used JMP Pro 13 (SAS Institute Inc., Cary, NC) to regress salinity with (a) genetic diversity and (b) the proportion of singleton haplotypes (calculated from the haplotype analysis described above). This latter approach was used to determine whether there was any influence of salinity on the proportion of rare haplotypes in a population. Additionally, given the variability in salinity within these systems, we also explored whether there was any relationship between the standard deviation of salinity and genetic diversity.

In addition, Primer 7.0.13 was used to construct rarefaction and extrapolation curves of haplotype diversity in order to determine the accumulation of haplotypes with sample size, the expected haplotype richness in each estuary, and the number of samples that would produce an asymptote in haplotype richness (extrapolation). Nonparametric estimators have been found useful in a number of studies for predicting the eventual asymptote in richness of a particular population (Gotelli & Colwell, 2001) and do so by including the effects of rare (or singleton) species/haplotypes (Witman, Etter & Smith, 2004; Chao, 2005; Blakeslee, Byers & Lesser, 2008; Blakeslee et al., 2012). A clearly asymptoting accumulation curve indicates complete capture of the total richness in a population (Gotelli & Colwell, 2001), thus estimator curves and accumulation curves that converge on the same asymptote can be very useful in determining whether there is adequate sampling in a population or region, or whether more sampling would reveal additional species/haplotypes (Walther & Morand, 1998; Blakeslee, Byers & Lesser, 2008).

For our second analysis focused on goby biogeography, we performed an AMOVA to explore differences at the subregional level and also at the larger regional level, specifically between the Atlantic and Gulf of Mexico regions. We again used PopArt to graphically create haplotype networks, using incidence-based data from Mila et al. (2017), Van Tassell et al. (2015), and our new North Carolina populations. In this analysis, we also included five sequences from a non-native population in Germany (Van Tassell et al., 2015) to identify if a source for the non-native population could be revealed within the overall dataset.

Results

North Carolina estuaries

In our North Carolina dataset (Fig. 1; Table 1), we uncovered a total of 30 previously undescribed haplotypes. Among these haplotypes, 74% of our 155 individuals (from four sites in the Pamlico estuary and five sites in the Neuse estuary, and also a site from Bogue Sound) were found to share one dominant haplotype (HAP5). The dominance of this haplotype and the connections among it and the other haplotypes in the estuaries can be observed in the haplotype network of North Carolina populations (Fig. 2). For our two major estuaries (Pamlico, n = 88, and Neuse, n = 59), this haplotype (HAP5) was slightly more frequent in the Pamlico (78%) versus the Neuse (68%). At the haplotype level, most haplotypes were singleton occurrences: in the Pamlico, 60% of the haplotypes were singletons, and in the Neuse, 74% were singletons. When comparing between the Neuse and Pamlico sites, just 4 haplotypes (14%) were shared between the estuaries; thus at the haplotype level, there was much less overlap between them. However, at the individual level, 86% of the Neuse individuals shared haplotypes with the Pamlico, and 76% of the Pamlico individuals shared haplotypes with the Neuse. This was supported by non-significant differentiation in the AMOVA comparing these two estuaries (FCT = −0.00149; p = 0.44282). It is important to note, however, that most of the sharing between estuaries and among populations occurred with the dominant haplotype, HAP5. In addition to the dominant haplotype (HAP5), the second most frequent haplotype (HAP27) was also comprised of individuals from both estuaries, as well as Bogue Sound.

Figure 2 Haplotype network—North Carolina.

Subregions include: NC_P (Pamlico), NC_N (Neuse), NC_B (Bogue Sound). The size of the circle is representative of the number of occurrences for each haplotype (see key).

At the population level (Fig. 3), the following sites were significantly (or nearly significantly) different from one another in pairwise ϕST analyses after accounting for multiple pairwise comparisons (Bonferroni correction, p = 0.005): MLC and CQC (p < 0.001); MLC and HPC (p = 0.009); and NCL and HPC (p = 0.009). In addition, Table 1 lists the ratio of observed vs. expected haplotypes vis-à-vis the number of fish sampled at each site. GSC and CQC were found to have the greatest genetic diversity, with the number of expected haplotypes predicted to be roughly three times greater than what was actually observed. Both of these sites are classified as low in salinity (Table 1). Fish from all other sites (MLC, NCL, WRC, FSL, MTP, POC, CDI, HPC) occupied the full range of salinity (low-medium-high) and demonstrated between one and two times the expected number of haplotypes compared to the number sampled. At the estuary level (Figs. 4A–4B), eight times more haplotypes were predicted for the Neuse (n = 160) in the extrapolation curve than were actually detected (n = 20), and for the Pamlico ∼5 times more haplotypes were predicted (n = 80) than were detected (n = 15). Comparing between the estuaries, the Neuse was ∼2.5 times greater in predicted diversity than the Pamlico. Such diversity differences between the estuaries were also supported by a Shannon diversity test, finding H′= 1.55 for the Neuse and H′= 1.06 for the Pamlico.

Figure 3 MDS plot of North Carolina populations.

Pairwise FST data were analyzed using a resemblance matrix and plotted using nMDS. Points closer together are more similar in their haplotype frequencies than those more distant. Fish sampled from sites along the Pamlico River appear as triangles (GSC, MLC, NCL, WRC), and fish sampled from the Neuse appear as circles (FSL, CQC, MTP, POC). Fish from Bogue Sound appear as a square (HPC).

Figure 4 Rarefaction and extrapolation curves.

The upper graph represents accumulation (SOBS) and estimator curves of haplotypes in the two North Carolina estuaries, scaled by the number of individuals. The lower graph extrapolates an asymptote based on the number of unique haplotypes sampled from fish within each estuary.

In explorations of salinity and genetic diversity/proportion of singleton haplotypes, we found no significant correlations for the following regressions: genetic diversity and salinity (R2 = 0.004; p = 0.873; Fig. 5A), the proportion of singleton haplotypes and average salinity (R2 = 0.070; p = 0.490; Fig. 5B), and the proportion of singleton haplotypes and salinity standard deviation (R2 = 0.187; p = 0.245; Fig. 5D). At the population level, there was seemingly little influence of salinity and genetic relatedness. For example, two spatially similar sites (in terms of pairwise ϕSTs), CDI and NCL, had salinities that were separated on average by 6 ppt, and these sites were also found in different estuaries. However, there was a significant positive correlation between the standard deviation of salinity and genetic diversity (R2 = 0.500; p = 0.034; Fig. 5C). No correlations were found when estuaries were analyzed separately, nor did non-linear regressions improve fit and significance in any of these analyses, except in the case of the second polynomial for the proportion of singleton haplotypes and salinity standard deviation (R2 = 0.792; p = 0.009) (Table S3).

Figure 5 Salinity and genetic diversity.

Dark blue diamonds = Pamlico; light blue circles = Neuse. Salinity was averaged across four time intervals and regressed with (A) genetic diversity and (B) proportion of singleton haplotypes. Salinity standard deviations are also plotted for genetic diversity (C) and proportion of singleton haplotypes (D).

Biogeographic comparison

In our second analysis (Fig. 6), we combined G. bosc sequences from the public sequence repository, GenBank, with our own North Carolina estuary samples. For this analysis, we explored regional differentiation across North America, including the mid-Atlantic (New York and Virginia), North Carolina (Neuse, Pamlico, Bogue), South Carolina, Atlantic Florida, Florida Gulf, Florida panhandle, and other Gulf states (Louisiana, Mississippi, and Texas). In addition, we included five sequences found in a non-native population in Germany (Weser estuary). Except for North Carolina, these samples came from two studies: (Mila et al., 2017) and (Van Tassell et al., 2015). Among all these samples, a total of 88 haplotypes were found. These haplotypes were significantly regionally differentiated (p < 0.001), with North Carolina falling in with the southeast and mid-Atlantic samples. When comparing Atlantic versus Gulf of Mexico sites, there was significant differentiation between the two major regions (p < 0.001) (Fig. 6).

Figure 6 Haplotype network of Atlantic and Gulf of Mexico populations.

Subregions included the following: MIDATL (Mid Atlantic, including New York and Virginia), NC_P (NC Pamlico), NC_N (NC Neuse), NC_B (NC Bogue), SC (South Carolina), FLA (Atlantic Florida, including Jacksonville (JAFL) and Indian River (IRFL), FLG (Gulf Florida, including Cedar Key (CKFL) and Tampa Bay (TBFL)), FLP (Florida panhandle, including Apalachicola (APFL) and Destin (DEFL)), GOM (Gulf of Mexico, including Empire, Louisiana (EMLA), Ocean Springs, Mississippi (OSMS), and Galveston, Texas (GATX)), and GERM (Weser Estuary, Germany). This figure represents an incidence-based TCS network analysis of all haplotypes and their connections throughout the sampled region in the Atlantic and Gulf of Mexico. Much of the data comes from Mila et al. (2017) (acquired from GenBank) and also our recent data from North Carolina estuaries.

Introduced German population

At this point in time, any comparison with the non-native population (Weser estuary, Germany) must be taken with caution, as there are just a few (n = 5) representative sequences from this region (Van Tassell et al., 2015). However, in the haplotype analysis (see red coloration in Fig. 6) all five of those sequences aligned within the Atlantic network, with two individuals sharing haplotypes with Atlantic Florida and the mid-Atlantic. There was no evidence for a connection to the Gulf of Mexico. Additional data from the non-native region could help pinpoint a more specific origin for this non-native population.

Discussion

G. bosc is an important member of estuarine communities, but much remains unknown about the dispersal potential of this species, especially at smaller scales, and how it may influence gene flow among populations within a habitat mosaic, or among estuaries. Recent genetic work has focused on broader questions of phylogenetic relationships among the genus Gobiosoma (Van Tassell et al., 2015) and large-scale differentiation in widespread populations of G. bosc (Mila et al., 2017). Our study is the first to focus on smaller-scale patterns of connectivity and gene flow in naked gobies using a major estuarine system as our focal region. Our study also contributes previously undocumented sequence data from North Carolina populations of G. bosc, which will help in further resolving questions related to gene flow in this species at both the local and biogeographic scales. In addition, we incorporated a salinity gradient into our study design, as previous research has suggested that salinity is an important abiotic factor in the life history of this fish. To the contrary, we found that salinity alone was not a major predictor of genetic diversity in this species (with the exception of salinity variability), even though reports in the literature find naked gobies to be most abundant in moderate salinity habitats. In the sections that follow, we expand upon these findings and discuss their implications.

Genetic diversity, gene flow, and connectivity

Spatially-structured populations are linked by the dispersal of individual organisms (Cote et al., 2010). Dispersal itself is a fundamental life-history trait (Schludermann et al., 2012), and connectivity between distant groups is a major driver of population dynamics (Bignami et al., 2013). Moreover, the availability of stable epibenthic substrate is crucial for maintaining populations of estuarine fishes like G. bosc as well as other organisms (Allen & Barker, 1990; Shima & Swearer, 2009; Gain et al., 2017). In North Carolina, gene flow was broadly distributed across G. bosc populations between the Pamlico and Neuse estuaries (Fig. 2), and genetic diversity was quite high. The latter is especially reflected in the proportion of singleton haplotypes within populations (Table 1), which reached as high as 75% in one Neuse population and was close to 50% when averaged across all nine populations. Such a preponderance of singleton haplotypes suggests a significant amount of genetic diversity remains unaccounted for in this system. In fact, rarefaction and extrapolation curves (Figs. 4A–4B) suggest upwards of 1000s of individuals would need to be sampled to produce an asymptote; in turn, greater sampling would produce significantly more haplotypes in each estuary than were initially detected.

In addition to demonstrating high diversity and a large number of singleton haplotypes (just a couple mutation steps away from the dominant haplotype), the star-like pattern of the haplotype network is also a well-known signature of populations that have undergone recent expansion (e.g., Mila et al., 2000; Fratini & Vannini, 2002). For example, star-like patterns (together with other molecular analyses; e.g., Fu’s Fs and Tajima’s D) detected in the haplotype network of the protist Plasmodium falcipram (which infects African mosquitos) demonstrated clear population expansions for the parasite (Joy et al., 2003). In a marine example, star-like patterns were observed in haplotype networks of the spiny lobster (Palinurus gilchristi) in South Africa, suggesting a recent bottleneck and/or population expansion (Tolley et al., 2005). Specific to G. bosc, there is little understanding of this fish’s ecological and demographic history in this region, and as a result, any explanations of the mechanisms responsible for population expansion are inherently speculative. However, as winds are the main driver of currents and water level in the Pamlico and Neuse estuaries (Luettich et al., 2002; Whipple, Luettich & Seim, 2006), larval G. bosc would be highly susceptible to wind-driven dispersal while remaining planktonic in the water column. Given the severe storms that frequent the region (and also coincide with seasonal spawning activity of G. bosc), our evidence for a possible population expansion could be a response to one or more of the 35 tropical cyclones to affect coastal North Carolina over the past two decades (Paerl et al., 2018), or perhaps could be due to recent temperature and climactic changes in the region (Harley et al., 2006). A greater understanding of the interaction between the biology of this species and the multiple abiotic factors shaping its distribution, reproduction, and gene flow in the past and present is therefore needed.

Naked gobies also demonstrate extensive gene flow between and among populations in both estuaries, but especially within the Pamlico River. For example, in the MDS plot of pairwise ϕSTs (Fig. 3), Pamlico sites are closer together spatially than Neuse sites. The location of our sample sites along each river, as well as the topography of the two rivers themselves, could be influential in differentially affecting gene flow in these rives. Most of our sites were located on creeks that were tributary to the main stem of either river. Although these sites were all positioned within 1 km of the river, some creeks would be subject to more flushing than others, while more hydrodynamically-isolated areas would favor greater larval retention. In addition, the orientation of the Pamlico itself is straighter (Fig. 1) and thus may be more favorable to wind-driven dispersal relative to the Neuse, the latter demonstrating an orthogonal bend between the upper and lower parts of the estuary. In some areas of the Neuse estuary, this may possibly lend itself to greater larval retention, greater isolation, and potentially more locally adapted populations. Retention zones have been detected in numerous marine systems and are highly important to the genetic structure and diversity exhibited among populations of organisms connected by larval dispersal (Palumbi, 1994; Pringle & Wares, 2007; Pringle et al., 2011). Larval dispersal is especially important in organisms that are much less mobile (or sessile) as adults (Sotka et al., 2004). While fish tend to be highly mobile during both larval and adult stages, fish species like G. bosc that are more associated with the benthos and have strict habitat requirements may be more influenced by retention zones that will influence gene flow and genetic structure within a region. In future studies, it would be important to map the availability of preferred settlement habitat in each river to determine how well it corresponds to the observed distribution of haplotype frequencies among our sample locations.

The role of salinity and habitat preference

Fish were collected along a salinity gradient averaging 3 to 14 ppt (averaged across four time periods in 2017) (Table 1). Although G. bosc is a euryhaline species, salinity alone did not seem to affect the distribution of haplotype frequencies across our sample sites, even though the greatest abundance of larval naked gobies in plankton tows has been reported from mesohaline habitats (e.g., Dawson, 1966; Shenker et al., 1983). In our study, adult naked gobies were most abundant in sites ranging from 4–12 ppt. We therefore expected greater genetic diversity in fish collected from mesohaline sites because of their relative abundances. However, our results (Fig. 5) revealed few significant linear or non-linear trends with salinity (albeit our sampling only incorporated a single polyhaline site) with two exceptions: (1) we found a significant positive relationship between genetic diversity and salinity standard deviation (Fig. 5C), whereby sites with greater salinity variability possessed greater genetic variability; and (2) we found a significant positive relationship between the proportion of singleton haplotypes and salinity standard deviation (Table S3). Such an outcome could signify the influence of neutral processes like the effects of waves and currents as one moves upriver, or potentially more fine-scale genetic structure as a result of varying salinity (Beheregaray & Sunnucks, 2002). Alternatively, it may signal some kind of salinity adaptation occurring along salinity gradients in both estuaries. Though COI is generally treated as a neutral marker (i.e., it is widely used as a “barcoding” gene), this may not always be the case, particularly when linked to genes that are under greater selective pressures (Moritz & Cicero, 2004). Thus, differences in genetic variation along a salinity gradient may suggest an adaptive response (i.e., more variable salinity represents less stable conditions in a population); however, a genome-wide approach would be necessary to elucidate this potential influence on genetic variation in this system.

Further, previous studies have recognized the importance of habitat in structuring populations of G. bosc, but have also advocated for the major synergistic role that salinity plays for this fish species. Larval G. bosc approaching competence-to-settle are known to congregate near the downstream edges of oyster reefs, rubble, or artificial structures (Breitburg, 1989; Breitburg, 1991; Breitburg, Palmer & Loher, 1995). Adults are mostly sedentary and require hard substrate (e.g., oyster shell) for the attachment of egg masses, and it has been proposed that spawning is confined to the downriver portions of estuaries where this habitat is more common (Massmann, Norcross & Joseph, 1963; Shenker et al., 1983). Shenker et al. (1983) reported that the abundance and size of larval G. bosc increased with time in the upriver portions of the Patuxent River estuary in Maryland, and they speculated that larvae were selectively using flood tides to move upriver. Upriver displacement of larvae offers a low salinity refuge from predation and is common in other estuarine-dependent species like weakfish Cynoscio regalis (Lankford & Targett, 1994) and red drum Sciaenops ocellatus (Stewart & Scharf, 2008). However, the synchronous spawning movement of adult G. bosc seems highly unlikely given that the species is cryptic and adapted to a benthic lifestyle. Adult G. bosc are opportunistic in their choice of spawning substrate, and in lieu of oyster shell will use rubble, woody debris, or other material like discarded cans (Nero, 1976; Lehnert & Allen, 2002). Much like oyster shell, this habitat is often distributed haphazardly, and so population connectivity would depend on the quality, scale, and proximity of available habitat patches. G. bosc is adept at using such material, as evidenced by Miller & Guillory (1980), who extensively sampled a 155 km portion of the middle St. Johns River in Florida averaging 0.2 to 1.2 ppt. Based on the size and abundance of nearly 50,000 larval G. bosc, they unequivocally concluded that adults were spawning at all of their sample sites and that larvae were not being transported upriver. They also confirmed the presence of spawning adults and larvae in freshwater lakes and tributary streams of the St. Johns, which accords with our sampling of adult G. bosc in sites at the freshwater interface of the Pamlico River (<0.1 ppt). Altogether, the evidence from prior studies and our own suggests that gene flow in this species is mediated by multiple biotic and abiotic factors—principally among them the type of epibenthic substrate available as habitat, how the distribution of this habitat changes with salinity, and the location of suitable habitat relative to wind-forced circulation patterns.

Biogeography of US populations

Biogeographic breaks leading to genetic differentiation have been detected in a number of marine organisms, and North Carolina in particular is positioned at one of the sharpest marine thermal boundaries in the world (Pietrafesa, Janowitz & Wittman, 1985). As such, it functions as an important biogeographic break between temperate and sub-tropical regions. However, while we did not see a clear biogeographic break in Atlantic populations around North Carolina (as there was for the Gulf of Mexico; see discussion below), no shared haplotypes were detected between North Carolina and any other regions/subregions in the Atlantic—demonstrating that while there is a large amount of local gene flow within estuaries, there is much less gene flow at the regional level. Occasional stochastic weather events like tropical storms (AOML, AOML-NOAA, 2017) may move individuals (particularly larvae) beyond these local boundaries, but these movements appear rare. For example, Ross & Rohde (2004) report just a single record of an adult G. bosc collected from an offshore scallop bed, and there are occasional reports of juveniles collected off Beaufort, N.C., located close to the Atlantic Ocean (Hildebrand & Cable, 1938).

Interestingly, Atlantic populations of G. bosc demonstrated less noticeable differentiation among geographically spaced locations than within the Gulf of Mexico (i.e., Atlantic locations were separated by fewer sequence changes than were Gulf locations). For example, in the Gulf, geographic differentiation was much more apparent with a considerable break at Apalachicola Bay, sub-dividing the Florida panhandle in two (Mila et al., 2017). In the Atlantic, it may be that the Gulf Stream is playing a role in the greater genetic connectivity (albeit still subregionally differentiated) that was detected among Atlantic populations compared to the Gulf of Mexico populations. The Gulf Stream is known to shape dispersal patterns in many marine fauna. For example, it is thought to be responsible for the presence of Caribbean mesopelagics in the northern Sargasso Sea (Jahn, 1976), and some western Atlantic groups in the eastern Atlantic Azores archipelago (Avila et al., 2009). In addition, it also promotes gene flow among such widely distributed organisms as sea turtles (Blumenthal et al., 2009) and American and European eels (Kleckner & McCleave, 1982). These organisms all represent pelagic-spawning species whose larvae would be subject to long-distance transport, in contrast to G. bosc, which is an estuarine-resident organism. If the Gulf Stream does play a substantive role in gene flow among Atlantic populations of G. bosc, this might also help explain the importance of the tip of Florida representing a major biogeographic break. Off the southern tip of Florida in the Florida Straits, the Gulf Stream current flows east and is at its narrowest and strongest (Gula, Molemaker & McWilliams, 2015), which would likely reinforce divergence between Atlantic and Gulf populations at the regional scale.

Status of a non-native German population

Very little is known about the non-native population of naked gobies in the Weser estuary, Germany. Thiel, Scholle & Schulze (2012) reported that multiple individuals were collected in a stow net in 2009 by a commercial fishery vessel at a depth of between 11.0 and 14.3 m—unusually deep for this species. Without more information, it is unclear whether the population in Germany is more widespread or isolated to this particular estuary. Moreover, it is not possible at this point to determine a source location for this introduction, except that it appears likely to have come from an Atlantic source (i.e., in our network analysis, two Germany individuals were found to share haplotypes with an Atlantic Florida and a New York individual; Fig. 6). Given the distance between these two US Atlantic populations, this may suggest multiple introduction events from different source populations—a common occurrence among non-native species introduced via ballast water from shipping (e.g., Blakeslee et al., 2017). The Weser estuary is located near the border with the Netherlands and serves as an important commercial shipping hub, which led Thiel, Scholle & Schulze (2012) to speculate that G. bosc was introduced via ballast water. An additional introduction has previously been reported from the Orinico Delta in Venezuela (Lasso-Alcala, Lasso & Smith, 2005), which is also a major international shipping destination. In all probability, introduced populations of G. bosc are underreported owing to the species’ small size and cryptic nature.

Conclusions

We stand to learn a great deal from studying common fish like G. bosc that, notwithstanding their abundance, remain relatively understudied in the literature. Only recently, for example, was a comprehensive analysis published on the feeding ecology of this species (e.g., D’Aguillo, Harold & Darden, 2014), which addressed basic questions like diet and daily patterns in foraging activity. In the future, we intend to continue to address the limited understanding of the basic biology of this species by resolving questions of adult dispersal—in particular by studying the movement patterns of adult G. bosc that reside in specific habitat patches. The distribution of adults also appears to be related to shoreline exposure, as areas with significant fetch tend to have fewer naked gobies and a higher relative abundance of other species like skilletfish Gobiesox strumosus and striped blennies Chasmodes bosquianus (C Moore, pers. obs., 2018). While adult G. bosc are relatively cryptic and do not appear to stray far from complex habitat, this remains an untested assumption that could potentially inform our understanding of population connectivity in this species at the estuary level. Further resolution of the genetic connectivity among populations should also include genome-level markers (i.e., RAD-Seq), as it can be difficult to develop a detailed understanding of the magnitude of gene flow at smaller scales using only frequency-based approaches (Waples, 1998; Hellberg, 2009). Even so, our study provides an initial understanding of the importance of investigating genetic diversity and population structure at local and regional scales.

Supplemental Information

Table S1 Haplotype data and sequence information

In the table below, the first column represents the haplotype number (the actual sequence information for each haplotype follows this table). Columns 2–10 represent newly collected raw data from populations in North Carolina. These include the abundance of a particular haplotype per site. Columns 11–12 represent data extracted from Genbank that were part of Van Tassell et al. (2015). Columns 13–23 represent incidence data extracted from Genbank that were part of Mila et al. (2017).

Click here for additional data file.

Table S2 Pairwise FST values

Click here for additional data file.

Table S3 Regressions with salinity

Click here for additional data file.

Supplemental Information 1 Sequence data

Sequence data for North Carolina Populations, Atlantic and Gulf of Mexico Populations (Mila et al., 2017), and German Populations (Van Tassell et al., 2015) of G. bosc

Click here for additional data file.

Additional Information and Declarations

Competing Interests

Author Contributions

Animal Ethics

Field Study Permissions

Data Availability

The authors declare there are no competing interests.

Christopher S. Moore conceived and designed the experiments, analyzed the data, prepared figures and/or tables, authored or reviewed drafts of the paper, approved the final draft.

Matthew J. Ruocchio conceived and designed the experiments, performed the experiments, authored or reviewed drafts of the paper, approved the final draft.

April M.H. Blakeslee conceived and designed the experiments, performed the experiments, analyzed the data, contributed reagents/materials/analysis tools, prepared figures and/or tables, authored or reviewed drafts of the paper, approved the final draft.

The following information was supplied relating to ethical approvals (i.e., approving body and any reference numbers):

East Carolina University IACUC animal use protocol (#D346) provided full approval for this research.

The following information was supplied relating to field study approvals (i.e., approving body and any reference numbers):

Field experiments were approved by the North Carolina Division of Marine Fisheries (permit number: 706671).

The following information was supplied regarding data availability:

The raw data are provided in a Supplemental File. Sequence data for accession numbers MH680722–MH680751 can be found in Table S1.

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
