# Peer review of "Distribution and population structure in the naked goby Gobiosoma bosc (Perciformes: Gobiidae) along a salinity gradient in two western Atlantic estuaries"

_PeerJ, doi:10.7717/peerj.5380_

## Round 0.1 · original submission · Major Revisions

We now have 2 reviews back on your submission, and while both see the value of the submission, each also comments that the conclusions overstep the strength of the data. Each has multiple recommendations to improve the submission but come to slightly different conclusions about the seriousness of the issues. Certainly, comments about the lack of information regarding sample sizes and experimental design must be addressed in the revision. Even the more encouraging of the referees states that the authors are trying “a little hard to make something out of nothing” with some of the conclusions. PeerJ does not evaluate manuscripts on their story or impact, only on the scientific merit and validity of the results, and so there is no need to overstep the data and draw grandiose conclusions. I find myself in agreement with the referees that parts of the manuscript are highly speculative, particularly considering the unknown but overall small sample sizes per site. Both referees recommend toning down the conclusions and instead sticking to the main findings of the study. While I am not opposed to speculation, I do expect it to be clearly labeled as such. If you wish to speculate, please first make clear your conclusions that are well supported by the data and then go on to the speculative portion with some obvious transition such as “We hypothesize that… and future research will determine whether our hypothesis here is upheld.” At least by separating your conclusions and speculation it is more scientifically defensible and makes clear to referees and readers alike what you think is happening, but do not have the data to support.

Reviewer 1 ·

Basic reporting

The paper “Distribution and population structure in the naked goby Gobiosoma bosc (Perciformes: Gobiidae) along
a salinity gradient in two western Atlantic estuaries” describes the genetic population structure of the naked goby along two estuaries in North Carolina using the marker CO1 and places it in a larger geographic context using published CO1 sequences. The two main questions are whether the populations will show genetic structure at a small geographic scale (estuary) and if this structure correlates with the salinity gradient in the estuary.

The biggest weakness of the paper are:
1. We are missing information on the number of samples collected at each site.
2. Discussion points are in the results and introduction material is in the discussion
3. Significance of results are exaggerated
4. Some of the discussion is off topic or missing key points to discuss

Experimental design

Is there a reason why fin clips and marking the fish could not have been used instead of killing the fish? If so it should be in the methods.

In general: nowhere in the methods is it clear how many gobies you sampled per site and how many samples you use in the larger geographic comparison. You mention “a maximum of 10 gobies” but we don’t know exactly how many. This should be clearly shown in Table 1 so that we can compare the number of haplotypes to the number of fish sampled. This is a big flaw for this analysis!

L 176 “from five locations” should read “from 9 locations”.

L 189-190 “ten collecting units per river” There are only 4 sites in the Pamlico river, so that would be 8 collecting units.

L 208 to 214 – This should be in the introduction – why you think there should be a genetic gradient correlating to a salinity gradient.

L 227 What type of sequencing was carried out? Sanger? MiSeq? Please be more precise.

L229 Did you align them to a reference CO1 sequence? Did you clean the sequences you received? Please be more specific.

L 230 – extra closing bracket “New Zealand))”

L 243 – 252 This paragraph is confusing and presents too many different points – restructure. Where do you get the abundance data from????

L 257 Phist – Please indicate the table #
L258 nMDS – Please indicate the figure #
L 260 TCS network – please indicate which figure? Use “haplotype network” instead of TCS network throughout the paper.

L 261 New paragraph for salinity – “We used ARLEQUIN to obtain…”

L 267 New paragraph for haplotype accumulation

Validity of the findings

There are some parts of the results that read like the discussion. Please remove those sections and develop them in the discussion (see below).

The haplotypes shown in figure 1 are difficult to see (too small) and are not discussed in the results.

L298 –This conclusion should not be here but in the discussion and discussed with all the caveats. Also you haven’t shown if these differences were significant. This could be due to a difference in sampling, undersampling or could be random.

L300 The MDS results are not that clear and again should not be discussed in the results but in the discussion.

L329 – 351 – The title of the paper is about population structure along a salinity gradient – it seems that table S1 and Figure S1 are central to this and should be in the body of the paper. At least the salinity table.

L329. Remove “also”

Discussion

L 379 to 404: This entire paragraph belongs to the introduction. Please re-write presenting the key finding you would like to discuss in the sections you’ve defined (you do this around L 399)
L414: If you are going to make such a broad statement, make sure you have the literature backing it up, including the most recent ones. I think you will find that the state of the literature shows a much more complex story with examples of marine dispersal being dictated by small-scale factors. You don’t actually need to make a comparison with the marine realm here to make your point regarding estuarine fish.

L419 to 421 This sentence doesn’t make sense – If there is gene flow why would you expect to find barriers to gene flow? How can you find both gene flow and population structure? Please clarify. The star-like structure of Fig 3 with a lot of new haplotypes with few mutations between each of them suggests to me is that there may have been a recent population expansion in these estuaries where haplotypes are not in equilibrium. This needs to be discussed.

L425-428. Your point here doesn’t support the theory of barrier to gene flow. Also, cite references for each of your ideas - develop them in separate paragraphs (for e.g. idea that habitat patchiness will influence population structure).

L441-461 Your data never showed that there was a significant difference in shared diversity between the estuaries – the data doesn’t reveal “clear differences”. Either add analysis showing that the observed differences are statistically significant or review this paragraph.

L479-492 CO1 has been shown to not always be neutral. II think it is important to discuss in this section the possibility that you may be seeing adaptation to salinity (and not wave and currents.). If this population is in recent expansion and not in equilibrium, you could be seeing haplotypes variability due to adaptation to salinity. You miss the point in this paragraph. It needs to be reviewed.

L494 to 520 It is difficult to understand the point you are trying to make in this paragraph. The upriver migration of larvae doesn’t fit here and is not relevant to the point you are trying to make. Start the paragraph at L506 “Adult G.Bosc”… The point of complex habitat influencing genetic diversity is a valid one but gets lost. It would be good here to see citations of other work that shows that habitat patchiness can reduce gene flow and increase genetic structure.

Conclusion

The excerpt from Emily Dickinson is nice but doesn’t fit in the conclusion – takes away from the points you want to make.

Reviewer 2 ·

Basic reporting

The paper is a very well-written manuscript. The references are thorough and the background and context provided are very strong. The hypotheses are relevant and clearly stated, the structure of the manuscript is appropriate, and the tables and figures are of good quality. I have made some comments regarding the figure legends, as they are too large and much of the information there should be in the main text, not the legends.

Experimental design

The sampling design was described in adequate detail, and is appropriate for this type of study, however based on the findings of the study, it appears that many, many more samples would be needed to strongly support/reject their hypotheses, due to the large number of unique haplotypes and the tremendous amount of unsampled diversity still sitting out in the wild populations. Based on the rarefaction curves, I can't imagine how any study with a finite budget/timeline would be able to sample the number of individuals required to adequately summarize the patterns here. The estimates on the number of haplotypes out there is itself is a pretty cool finding. The authors make good use of the existing public data, especially by incorporating the invasive population from the phylogenetic study with the previous pop-gen dataset.

I do not see if/where the new sequences generated from this study were deposited. The authors should plan to make these data available on GenBank, noting the accession numbers in the methods. Space-filler numbers (eg. "Accession numbers XXXXX-XXXXX") are fine for review purposes until the paper gets published.

Validity of the findings

Overall I think that the main findings of the study are that:
1) there is no significant genetic structure within the two estuaries samples
2) there is no effect of salinity
3) The overall position of NC in the biogeographic picture is consistent with what one would expect based on prior results.

With regards to #1 and #2, I think that the authors may be a little guilty of trying a little hard to make something out of nothing, especially when looking at the relationship of salinity variability and genetic diversity. I think that the positive correlation that they found, if it is indeed real, has less to do with actual salinity itself (i.e. salinity is not really a driver for local adaption), and more to do with the fact that the areas with variable salinity are also those that are subject to mixing of water (and thus recruits) from different areas around the estuaries. I also think that the comparison of the % of individuals with shared haplotypes, or the number of singleton haplotypes, across the two bays is not all that meaningful, because there are SO many unsampled haplotypes. Trying to come up with biological reasons for differences in these two metrics is highly speculative, considering that with such small sample sizes we are barely scratching the surface of the true diversity in the system.

I recommend toning down or removing the talk of adaption in the abstract and elsewhere, and instead sticking to the main findings I listed above. While #'s 1 and 2 could be viewed as negative or inconclusive results, I think they are highly relevant for this system, and perhaps expected given the ecology of the organism.

Additional comments

I have made some additional comments and suggestions on the attached PDF as sticky-notes. These largely echo the comments above, but are more detailed in some cases.

Overall, I think the study is publishable pending some minor revisions, which relate to not overstating the findings given the limitations of the data.

Annotated reviews are not available for download in order to protect the identity of reviewers who chose to remain anonymous.

---

## Round 0.2 · accepted · Accept

Thank you for your detailed response to the referee comments. The most critical of the referees has re-evaluated the paper and I agree with them that the revisions have addressed the concerns raised by both referees in the first round. Therefore, I see no reason to prolong the process, and am happy to move the paper forward into production.

# Reviewer 1 ·

Basic reporting

Clear writing - the figures, tables and manuscript has been revised.

Experimental design

The authors have cleaned up the methods and clarified table 1.

Validity of the findings

The authors have reviewed the discussion according to the reviewers comments.